# Improving GIS-Based Landslide Susceptibility Assessments with Multi-temporal Remote Sensing and Machine Learning

**DOI:** 10.3390/s19173717

**Published:** 2019-08-27

**Authors:** Jhe-Syuan Lai, Fuan Tsai

**Affiliations:** 1Department of Civil Engineering, Feng Chia University, Taichung 40724, Taiwan; 2Department of Civil Engineering, National Central University, Taoyuan 32001, Taiwan; 3Center for Space and Remote Sensing Research, National Central University, Taoyuan 32001, Taiwan

**Keywords:** GIS, landslide susceptibility, machine learning, remote sensing, spatial analysis

## Abstract

This study developed a systematic approach with machine learning (ML) to apply the satellite remote sensing images, geographic information system (GIS) datasets, and spatial analysis for multi-temporal and event-based landslide susceptibility assessments at a regional scale. Random forests (RF) algorithm, one of the ML-based methods, was selected to construct the landslide susceptibility models. Different ratios of landslide and non-landslide samples were considered in the experiments. This study also employed a cost-sensitive analysis to adjust the decision boundary of the developed RF models with unbalanced sample ratios to improve the prediction results. Two strategies were investigated for model verification, namely space- and time-robustness. The space-robustness verification was designed for separating samples into training and examining data based on a single event or the same dataset. The time-robustness verification was designed for predicting subsequent landslide events by constructing a landslide susceptibility model based on a specific event or period. A total of 14 GIS-based landslide-related factors were used and derived from the spatial analyses. The developed landslide susceptibility models were tested in a watershed region in northern Taiwan with a landslide inventory of changes detected through multi-temporal satellite images and verified through field investigation. To further examine the developed models, the landslide susceptibility distributions of true occurrence samples and the generated landslide susceptibility maps were compared. The experiments demonstrated that the proposed method can provide more reasonable results, and the accuracies were found to be higher than 93% and 75% in most cases for space- and time-robustness verifications, respectively. In addition, the mapping results revealed that the multi-temporal models did not seem to be affected by the sample ratios included in the analyses.

## 1. Introduction

Landslide risk assessment and management are critical and extensive tasks that provide systematic and rigorous processes for slope engineering practice and management [1], making them important tools for addressing landslide hazards [2]. The framework of landslide risk assessment and management consists of several subtasks, such as susceptibility, hazard, and risk assessments. In addition, van Westen et al. [3,4] proposed a landslide risk assessment procedure that was mostly consistent with Dai et al. [2]. The term landslide susceptibility represents the likelihood (0 to 1) or degree (e.g., low, moderate, or high) of landslide occurrences in an area with given local terrain attributes [5]. Although landslide susceptibility assessment is a fundamental and essential task within the landslide risk assessment and management framework, there is no general consensus at present as to which is the best procedure and algorithm to evaluate it [6,7]. Therefore, landslide susceptibility assessment must be studied and explored further to improve results of the subsequent tasks.

With the development of computing power, geospatial data, and technologies, a growing number of studies have explored the potential of using machine learning (ML) based algorithms with geographic information system (GIS) datasets and remote sensing images for constructing landslide susceptibility models on the regional scale, such as decision trees (DT) [8,9,10], entropy- and evolution-based algorithms [11,12,13,14], fuzzy theory [15,16,17], neural network [12,18,19,20,21,22], neural-fuzzy systems [23,24], and support vector machines [20,21]. Random forests (RF) has received increased attention in the ML domain in recent years because of the following advantages: (1) Excellent accuracy [25], (2) processing speed [26,27], (3) few parameter settings [28,29], (4) availability of high-dimensional data analysis (e.g., hyperspectral image cubes) [30,31], and (5) insensitivity to imbalanced training data [32]. Nevertheless, the performance of RF for landslide susceptibility assessments has seldom been explored, compared with traditional methods such as Logistic regression.

Both landslide inventory and landslide-related factors are critical components for constructing regional landslide susceptibility models. In the modeling process, sampling ratio (or called proportion in this study) is an important setting resulting in the model performance. An assumption of equal sampling ratio is not reasonable in many real-world applications (e.g., diagnosis and landslide prediction). Heckmann et al. [33] indicated that the proportion of landslide to non-landslide samples for data-driven analysis (i.e., statistical and data mining or ML approaches) often ranges between 1:1 and 1:10. Berry and Linoff [34] also proposed an oversampling strategy for significantly imbalanced cases, where the ratio of occurrence to non-occurrence samples ranged between 10% and 40%. According to the above experiments, proportions of 1:1, 1:4 (i.e., 25%, the middle of the oversampling range), 1:7, and 1:10 with the corresponding landslide-related factors have been considered in this study for constructing GIS-based landslide susceptibility models. To address the unbalanced prediction results caused by unequal sampling ratios or spatial and temporal uncertainties, the model was constructed with an extremely false alarm or missing error (called over-fitting effect), and the decision boundary was adjusted using cost-sensitive analysis to improve the RF algorithm.

Traditionally, landslide area and susceptibility were evaluated by extracting remote sensing images and by using related techniques for a period (called multi-temporal landslide inventory and susceptibility assessment). The areas of landslide triggered by a single event can be detected automatically or semi-automatically because of increased access to remotely sensed images with high spatial and temporal resolutions [35,36,37,38,39,40,41,42]. Subsequently, landslide vulnerability and risk assessments can be made [43] based on the event-based landslide inventory mapping and susceptibility assessment [43,44].

Two strategies are generally used for model verification, namely testing space-robustness and testing time-robustness [45]. The former method separates the data sample based on a single event or the same dataset into training and check datasets and the latter method predicts (classify) later or different landslide events or periods, with a landslide susceptibility model constructed from a specific event or period. However, results on time-robustness verification are relatively few and less reliable compared with those related to space-robustness [6,46].

The main purpose of this study was to integrate GIS datasets and multi-temporal remote sensing inventory with ML to improve multi-temporal and event-based landslide susceptibility assessments of Shimen reservoir watershed in northern Taiwan and compare the results with those obtained using DT, Bayes network algorithm, and logistic regression algorithm, which are commonly used in the related field [6,10,46]. The definition of landslide in this study was the same as that in the study by Highland and Bobrowsky [47], that is, landslides include the downslope movement of soil, rock, and organic masses under the influence of gravity as well as the landform that results from such movement. In addition, the size of materials was ignored because this was a regional assessment. This study also explored the impact of sampling proportions during the modeling process. Furthermore, cost-sensitive analysis was applied to adjust the decision boundary for reduction of the over-fitting issue. All experimental results were evaluated through space- and time-robustness verifications.

## 2. Materials and Methods

### 2.1. Study Site and Data Preprocessing

The Shimen reservoir watershed (Figure 1), which is located in northern Taiwan, was selected as the study site because of numerous typical shallow landslide cases in this area. The total area of the study site covered approximately 764 km^2^ of the mountainous terrain. The average annual precipitation in the watershed was approximately 2500 mm/year. The elevation measured from the digital elevation model (DEM) ranged from 250 to 3500 m above sea level. More than 60% of the study area had a slope degree greater than 16°, and approximately 29% of the above area had a slope degree between 16° and 28°. The used DEM was generated by the Ministry of the Interior in Taiwan during 2002 to 2004. According to the geological and soil maps published by the Central Geological Survey of Taiwan, the watershed comprised nine geological categories (e.g., argillite, shale, quartzitic, and etc) and six primary soil types (e.g., clay, sand, loam and mixed soils). The collected soil map had some null values in the northern region, which were ignored in the landslide susceptibility assessments in this study. The primary land-cover in the watershed was forest but had limited agricultural activities near a few tribal areas. Detailed information regarding geology and soil as well as the spatial distribution of river, road, and fault within the study site has been mentioned in Tsai et al. [10].

The landslide inventory used in this study consisted of shallow landslide as identified with multi-temporal satellite remote sensing images and pixel-based change detection analysis from 2004 to 2008 [48]. Most of the detected landslides were also verified through field investigation. A summary of landslides detected after typhoons is listed in Table 1. They were rasterized into 10 m ×10 m cells to be compatible with other datasets for modeling. The deposition regions in the landslide samples were already eliminated using an empirical criterion [49], that is, the slope of landslide samples must be larger than 10°, otherwise they were considered the deposition area (non-landslide region). The spatial distribution of the landslide inventory can be also found in Tsai et al. [10].

A total of 14 landslide factors (Table 2) were collected or derived from the vector and raster GIS datasets. In addition, all factor data sets were resampled to a 10 m × 10 m raster format regardless of their original scale or spatial resolution to be compatible with the modeling system. For example, aspect, curvature, and slope factors were derived from the DEM. Furthermore, the vegetative factor normalized difference vegetation index (NDVI) was derived from the multispectral SPOT-4 and SPOT-5 satellite images, which were acquitted before the typhoon events (second column in Table 1) during 2004 to 2008, and provided by the Center for Space and Remote Sensing Research in National Central University of Taiwan. The NDVI values acquired at different times can be used to quantitatively represent the multi-temporal vegetation characteristics of the study site. To address the NDVI variations of the same surface due to difference in topographic, radiometric, and atmospheric conditions, this study adopted pseudo invariant feature (PIF) normalization [50,51] and Minnaert topographic correction [52] to minimize temporal and spatial differences. To obtain the rainfall distributions, this study applied the commonly used algorithms of inverse distance weighting (IDW) and Kriging interpolation. These methods are used to generate event-by-event accumulative and maximum hourly rainfall maps based on the rainfall-gauge data. The non-landslide samples were randomly extracted, and the numbers of samples were equal to 1, 4, 7, and 10 times the landslide samples in this study. Then these landslide and non-landslide samples can be used to extract the corresponding landslide factors to produce datasets. The data preprocessing and sampling steps show in Figure 2 in detail.

### 2.2. Developed Machine Learning Based Model

A three-phase study was developed for development of multi-temporal and event-based landslide susceptibility models. In the first phase, RF-based landslide susceptibility models were constructed as shown in Modeling of Figure 2. In the second stage, the cost-sensitive analysis was used to avoid the over-fitting predictions as also shown in Modeling of Figure 2. Finally, the developed models were verified using space- and time-robustness strategies to generate the landslide susceptibility map as shown in Assessment of Figure 2 and Section 2.3.

This study employed the RF algorithm [53], which is one of the ML-supervised methods, to construct the multi-temporal and event-based landslide susceptibility models. The concept of RF was similar to the DT algorithm, a classical and popular approach in the related domains. Both methods apply the information gain (IG) measure or Gini index to evaluate the degree of impurity of landslide factors or variables. A landslide-related variable, which has a larger IG or Gini value, must be selected because it has a higher priority for construction of a conditional node, and further, this factor should be ignored in the next computation. After several iterations, a sequence of rules was constructed that can be used to classify other instances. Unlike DT, the RF algorithm randomly separates training data into many subsets using the bootstrap algorithm to build many trees for optimization. Therefore, more favorable results can be obtained using the RF classifier than using the DT algorithm. Discrete (or nominal) and continuous (or numeric) formats are commonly used in the geospatial field. For nominal data, entropy theory is used to compute IG [10], as described in Equations (1)–(3), where E(A) indicates the entropy of all training data; p and n represent the numbers of positive (landslide) and negative (non-landslide) labels, respectively; E’(a) and v indicate the entropy and subset amounts of a specific landslide factor, respectively; E(a_j_) is the entropy of the subset in a specific landslide factor computed by Equation (1); and IG(a) means the IG of a specific landslide factor. For numeric data, the Gini index was applied to evaluate the IG [10] as described in Equations (4) and (5), where C indicates a segmented point for a specific landslide factor used to divide numeric data into two parts and N_1_ and N_2_ represent the numbers of a ≤ C and a > C, respectively.
(1)$$E\left(A\right)\;=\;-\frac{p}{p+n}\log_{2}\frac{p}{p+n}-\frac{n}{p+n}\log_{2}\frac{n}{p+n}$$
(2)$$E^{\prime \left(a\right)}=\;\overset{v}{\underset{i=1}{\sum }}\frac{p_{i}+n_{i}}{p+n}E\left(a_{i}\right)$$
(3)$$\mathrm{IG}\left(a\right)\;=\;E\left(A\right)-E\prime \left(a\right)$$
(4)$$\mathrm{Gini}\left(a\leq C\;\mathrm{or}\;a>C\right)\;=\;1-\overset{m}{\underset{i\;=\;1}{\sum }}\frac{n_{i}}{N}$$
(5)$$\mathrm{IG}\left(a,C\right)\;=\;\frac{N_{1}}{N}\mathrm{Gini}\left(a\leq C\right)+\frac{N_{2}}{N}\mathrm{Gini}(a>C)$$

According to cost matrix, the cost-sensitive analysis or learning can be used to adjust the decision boundary of a developed model, which reclassifies training samples to reduce and to balance the omission or commission error of certain classes [54,55,56]. The size of the cost matrix is equal to that of the confusion matrix. A confusion matrix consists of true positive (TP), false negative (FN), false positive (FP), and true negative (TN). In the cost matrix, diagonal values (i.e., TN and TP) represent the cost of correct classification, and remainder values (i.e., FN and FP) describe the misclassification costs between various classes. In general, the costs of diagonal and other elements are set to 0 and 1, respectively. The decision boundary can be changed by adjusting the cost of incorrect elements to include or exclude more samples for balancing the classification result of a certain class. However, a positive or negative effect on the results of other classes might occur. To balance the false alarm and missing rates and to maintain the reliability and veracity of the overall results, different costs were tested for the target class in this study. More precisely, the optimal prediction (R) for an instance x in class i can be expressed as in Equation (6), where P(j|x) is the probability for estimation of classification of an instance into class j and C represents the cost. The optimal prediction is class 1 (landslide) if and only if the expected cost of this prediction is less than or equal to the expected cost of predicting class 0 (non-landslide), as presented in Equation (7). Given p = P(1|x) and C_00_ = C_11_ = 0, Equation (7) is equivalent to Equation (8), and a threshold p* can be defined as Equation (10) based on Equation (9), which classifies an instance x as positive if the P(1|x) is larger than or equal to the threshold [57].
(6)$$R\left(i|x\right)=\underset{j}{\sum }P\left(j|x\right)C_{\mathrm{ji}}$$
(7)$$P\left(0|x\right)C_{10}+P\left(1|x\right)C_{11}\leq P\left(0|x\right)C_{00}+P\left(1|x\right)C_{01}$$
(8)$$\left(1-p\right)C_{10}\leq {p\;C}_{01}$$
(9)$$\left(1-p^{*}\right)C_{10}\leq p^{*}C_{01}$$
(10)$$p^{*}\;=\;\frac{C_{10}}{C_{10}+C_{01}}$$

### 2.3. Verification and Mapping

To verify the constructed landslide susceptibility models, the quantitative results of space-robustness and time-robustness verifications, which compare the output and reference labels (e.g., landslide or occurrence and non-landslide or non-occurrence) of the check or prediction datasets as identified by a threshold, are derived from the confusion matrix. In most cases, a sample is assigned into the non-landslide class when the landslide likelihood is smaller than 0.5, otherwise, the sample is not classified the non-occurrence label (landslide). The threshold was equal to p* after cost-sensitive analysis was conducted. Further, a landslide susceptibility map can be generated instance by instance based on the landslide likelihoods or output categories (e.g., low, medium and high susceptibilities) of the constructed model, if the verified results are acceptable.

This study adopted four commonly used quantitative indices for verification, namely overall accuracy (OA), precision (also called as the user’s accuracy, UA), recall (also called as the producer’s accuracy, PA), and area under the receiver operating characteristic (ROC) curve. The PA and UA as described in Equations (11) and (12) can reflect the errors class by class [10], where M is a component in a confusion matrix, Nc represents the number of classes, and i and j indicate the indices for columns and rows in a confusion matrix, respectively. Moreover, 1-recall and 1-precision can reflect the omission (missing) and commission (false alarm) errors, respectively. The OA presents the percentage of correctly classified samples [10], as described in Equation (13). The area under the curve (AUC) for ROC was also commonly used for evaluation of landslide susceptibility models [12,15,58,59,60], and generally, the value ranged from 0.5 to 1. A larger AUC represents higher reliability.
(11)$${\mathrm{PA}}_{j}\;=\;\frac{M_{\mathrm{jj}}}{\sum_{i=1}^{N_{c}}M_{\mathrm{ij}}}=\frac{M_{\mathrm{jj}}}{M_{+j}}$$
(12)$${\mathrm{UA}}_{i}=\frac{M_{\mathrm{ii}}}{\sum_{j=1}^{N_{c}}M_{\mathrm{ij}}}=\frac{M_{\mathrm{ii}}}{M_{i+}}$$
(13)$$\mathrm{OA}=\frac{\sum_{i=1}^{N_{c}}M_{\mathrm{ii}}}{\sum_{i=1}^{N_{c}}\sum_{j=1}^{N_{c}}M_{\mathrm{ij}}}=\frac{M_{\mathrm{diag}}}{M_{\mathrm{total}}}$$

## 3. Results

The modeling process and accuracy assessments in this study were performed using the WEKA platform (http://www.cs.waikato.ac.nz/ml/weka/), which is a free and open-source tool. The main parameter for RF-based computation was the number of trees (Ntree). Du et al. [26] mentioned that 10–200 Ntree did not have any influence on the results, but an increase in Ntree increased the computation loading. Therefore, in this study, 100 trees were used, which was suggested by WEKA, for executing RF-based landslide susceptibility assessments. The developed models were compared with the commonly used methods, namely DT, Bayes network, and logistic regression.

### 3.1. Multitemporal Landslide Susceptibility Assessments

The multi-temporal dataset used in the study included all landslide samples extracted after typhoon events between 2004 and 2008. The non-landslide samples of the same period were randomly extracted, and the numbers of samples were equal to 1, 4, 7, and 10 times the landslide samples in the test cases. These samples can be used to extract the corresponding landslide factors to produce multi-temporal datasets. For modeling, two-thirds of the samples from 2004 to 2007 (referred to as training data) were randomly selected from the multi-temporal dataset to construct the models, and the remainder (referred to as check data) were used for space-robustness verification. The constructed models were also used to predict the time-robustness verification of samples from 2008. Two prediction datasets were used in this study, namely multiple- and single-event samples. The multiple-event dataset indicated that the samples included all instances of 2008. On the contrary, the single-event dataset indicated that the collected records were extracted from a single typhoon event.

#### 3.1.1. Space-robustness Verification

The quantitative evaluations of space-robustness verification are presented in Figure 3. The OA may not fully represent the overall results of the unbalanced sample proportion cases, especially when the large sample class is easily detected, leading to an optimistic evaluation. Thus, the OA measure was only applied in equal sample proportion cases. Three important findings are illustrated in Figure 3. First, the space-robustness verification results revealed that the constructed models can provide flexible or acceptable results in all cases. In other words, the patterns of landslide occurrence from 2004 to 2007 can be described effectively. Second, in cases of unequally proportioned samples, the deterioration was observed in the landslide detection results obtained using the Bayes network and logistic regression algorithms, but the RF-based results were stable. More precisely, the Bayes network—and logistic regression–based models had lower UA and PA accuracies because of the increase in the unbalanced sample proportions. Third, in terms of the modeling performances, the RF-based results outperformed others in all cases.

#### 3.1.2. Time-Robustness Verification with Multiple-Event Samples

The prediction results of quantitative evaluation of the multiple-event samples of 2008 dataset are presented in Figure 4. The accuracies were significantly lower than those for the space-robustness results. Comparing Figure 4 with Figure 3, two points should be emphasized. First, although the landslide’s PA was slightly lower in the equal sample proportion case, the prediction capability based on the logistic regression algorithm yielded acceptable results. The RF-based model had higher commission and omission errors in terms of non-landslide and landslide detection, but the PA and UA accuracies were the highest for detection of the non-occurrence and occurrence classes in this case. The performance of the Bayes network algorithm was the lowest. Second, in the unequal sample proportion cases, the non-landslide detection results were improved because of the increase in the number of samples. However, although the logistic regression algorithm provided acceptable UA accuracies in all cases, a smaller number of landslide samples in the modeling process can cause a decrease in occurrence detection.

To address this issue, cost-sensitive analysis was employed to adjust the decision boundary during the classification-based modeling. The confusion matrices and quantitative accuracies for predicting the 2008 dataset were calculated using the RF algorithm after adjusting for the decision boundary. The results of the cost-sensitive analysis for different sample proportions are illustrated in Figure 5. The constructed model without cost-sensitive analysis (i.e., cost is equal to 1) had a high FN and low FP, resulting in unbalanced predictions and high missing and low false alarms. To address this issue and to reduce the problem of FN misclassification, the FN cost was increased for re-prediction of the 2008 dataset. A decrease in the TN and FN assignments was observed in the equal sample proportion case (Figure 5a). The trends of FP and TP were obtained from the low cost. In addition, a cross over between TN and TP as well as FN and FP occurred when the cost was 50. Figure 5b illustrates the optimal quantitative result, which was obtained at a cost of 50, where all indices were close to 0.8. For the unbalanced sample cases, an increase in TN values was observed with an increase in the number of non-landslide samples (Figure 5c,e,g). A decreasing trend for TN and FN and an increasing trend for FP and TP were observed with lower costs. A trade-off between landslide omission (missing) and commission (false alarm) errors in the unbalanced sample proportion cases is presented in Figure 5d,f,h. In the cases of unequal sample proportions, the results obtained based on costs of 100, 500, 1000, and 3000 (Figure 5d,f,h) exhibited higher performances, but produced unbalanced predictions between landslide’s omission and commission errors.

Same procedures of adjusting decision boundary were applied for the DT, Bayes network, and logistic regression algorithms. Further, Figure 6 compares the representative results (i.e., arrows in Figure 5) of all tests. Increasing PAs in unequal sample ratio cases was more critical than improving UAs in this study because landslide susceptibility assessment was the preceding part of the landslide risk assessment and management framework. Reducing commission errors (false alarm) can be expected through further tasks (e.g., hazard and risk assessments). Therefore, for the unequal sample ratio cases, the arrows in Figure 5 with lower omission errors (missing) were selected as the representative results for the comparison. On the basis of Figure 6, it appeared that the RF method with cost-sensitive analysis outperformed the other methods. In the 1:1 case, the RF model seemed to perform the best, considering that all the accuracy indices were larger than 0.79. The trade-off between the landslide’s UA and PA or preserving of the PA was also observed in the unbalanced sample proportion cases. Overall, RF models with cost-sensitive analysis can predict the unbalanced samples most effectively, although a certain degree of false alarms exists. To understand the interaction between the occurrence samples and the models, susceptibility distributions of true occurrence samples were illustrated with histograms in this study. Five susceptibility intervals were considered, namely very high (>80%), high (60%–80%), medium to high (40%–60%), medium to low (20%–40%), and low (<20%), where the higher likelihoods represented higher landslide occurrence potential. Ideally, all susceptibilities of the true landslide occurrence samples should be close to 100%, indicating true occurrence. However, the landslide susceptibility distributions obtained for the 2008 dataset on the basis of the model using the original RF algorithm were much lower than 100% (Table 3). These results also indicated that performing cost-sensitive analysis resulted in more reasonable landslide susceptibility results.

#### 3.1.3. Time-robustness Verification with Single-event Samples

The scenario discussed in this section is similar to that discussed in Section 3.1.2, but each prediction dataset consists of a single typhoon event. This type of verification was designed to explore the event-based prediction capability of multi-temporal landslide susceptibility models. The models were employed to predict landslides associated with Typhoon Fung-wong, Sinlaku, and Jangmi, which occurred in 2008, with different numbers of non-landslide samples. Similar procedures were applied to datasets from 2004 to 2007 for time-robustness verification of the models based on the abovementioned single-event samples. Representative results for prediction for Typhoon Fung-wong, Sinlaku, and Jangmi obtained using different algorithms are presented in Figure 7, Figure 8 and Figure 9. Examination of these figures revealed that the results produced by the RF method with cost-sensitive analysis were more favourable in most cases. In the 1:1 case, all indices obtained with the RF model for prediction of Typhoon Fung-wong, Sinlaku, and Jangmi were larger than 0.74, 0.73, and 0.82, respectively. A trade-off between the landslide UA and PA or keeping of the landslide PA was also observed in the unbalanced sample proportion cases.

This study also examined the landslide susceptibility distributions of the true occurrence samples for Typhoon Fung-wong, Sinlaku, and Jangmi based on the RF algorithm with cost-sensitive analysis (Table 4). Table 4 reveals that the developed models produced reasonable landslide susceptibility values for the true occurrence samples.

#### 3.1.4. Susceptibility Mapping

The multi-temporal landslide susceptibility maps generated from the developed models are presented in Figure 10. The RF models constructed without cost-sensitive analysis were too conservative (close to training data of occurrence area) and led to large omission errors. By contrast, the RF models with cost-sensitive analysis produced more stable and reasonable landslide susceptibility maps.

### 3.2. Event-Based Landslide Susceptibility Assessments

To ensure that there were sufficient samples to predict other events, the Typhoon Aere, Matsa, and Sinlaku samples, with the largest numbers of samples (Table 1), were selected to construct the landslide susceptibility models for space- and time-robustness verifications. Two-thirds of samples of each typhoon event were randomly extracted to develop the models, and the remainder samples were the check data. For time-robustness verification, one of the constructed models was employed to predict the other two events. The RF, DT, Bayes network algorithm, and logistic regression algorithm were also integrated with cost-sensitive analysis during the modeling process. In addition, sample ratios of 1:1, 1:4, 1:7, and 1:10 were considered for landslide and non-landslide classes.

#### 3.2.1. Space-Robustness Verification

Procedures similar to those discussed in Section 3.1.1 were applied to the selected typhoon events. The evaluation results are presented in Figure 11. Three important points were observed. First, the results in the equal sample proportion cases presented accuracies greater than 0.85. Second, there was an increase in the number of landslide commission (Typhoon Aere) and omission (Typhoon Aere and Sinlaku) errors in the Bayes network and logistic regression results for some typhoons in the unbalanced sample proportion cases. Third, the results from the RF- and DT-based models remained stable in all cases, especially those of the RF algorithm–based model.

#### 3.2.2. Time-robustness Verification

A procedure similar to that detailed in Section 3.1.3 was applied for Typhoon Matsa, Sinlaku, and Aere predictions. An overall comparison of the best prediction results for Typhoon Matsa, Sinlaku, and Aere is presented in Table 5. On the basis of this table, the accuracies were higher than 0.7 in the 1:1 cases, 0.63 in the 1:4 cases, 0.63 in the 1:7 cases, and 0.6 in the 1:10 cases, except for the landslide UAs of the unequal sample proportion cases. The results also indicated that the RF algorithm with cost-sensitive analysis performed better than the other algorithms. The evaluated landslide susceptibility results of the true occurrence samples for RF models with cost-sensitive analysis are presented in Table 6. Most of the true landslide samples were correctly labeled with high susceptibility.

#### 3.2.3. Susceptibility Mapping

The event-based landslide susceptibility maps generated based on the constructed models and Typhoon Sinlaku sample, as an example, are presented in Figure 12. The figures reveal that the equal sample ratio produced reasonable spatial patterns for event-based landslide susceptibility modeling, whereas the extremely unbalanced ratios seemed to cause over-fitting of the models because these patterns were close to the training data of occurrence area.

## 4. Discussion

The purpose of constructing multi-temporal and event-based landslide susceptibility models was to explore the capability of developed procedures for spatial pattern description and temporal prediction. Experimental results based on space- and time-robustness verifications as well as mapping results demonstrated the feasibility and stability of the developed algorithm (i.e., RF with cost-sensitive analysis). The reason was probably that the bootstrap and optimization steps in the RF algorithms and decision boundary adjustments in the cost-sensitive analysis improved the performance because they can help reduce the problem of over-fitting. However, the extremely unbalanced sample ratio was not suggested for event-based modeling due to the serious over-fitting issue.

In terms of the landslide and non-landslide samples, it was assumed that for most algorithms, misclassification errors have an equal cost. However, in many real-world applications, this assumption is not true. In this study, the results of time-robustness verification of unequal sample proportion cases and quantitative evaluation without cost-sensitive analysis revealed that the FN errors (i.e., missing or omission rates) were more serious than FP errors (i.e., false alarm or commission rates). This result was consistent with a previous study by Desai and Jadav [54]. From a statistical point of view, the low accuracies suggested a significant disagreement between the constructed models and prediction datasets. Two scenarios may have caused prediction errors. First, the predicted landslide samples revealed that similar conditions occurred in the past, but these locations in the prediction data were presently stable. Second, some landslide areas may have been classified into the non-landslide class because the models did not have similar occurrence situations derived from previous experience (training data). These may be the reasons for accuracies lower than those in the space-robustness verification.

The effectiveness of cost-sensitive analysis was explored to deal with landslide susceptibility assessments with balanced and imbalanced class distributions. Reducing omission (missing) errors in unequal sample ratio cases was more important than decreasing commission (false alarm) errors in this study because landslide susceptibility assessment was a preceding part of landslide risk assessment and management framework. Reducing commission errors was expected through further tasks. The experiments demonstrated that the accuracies of the developed models were better than 93% and 75% in most cases for space- and time-robustness verifications, respectively. It has been demonstrated that elaborate back analysis of past landslides gives insight of critical factors affecting their triggering [61,62]. Thus, back analysis on a number of observed past landslides of the region considered may give insight of critical factors affecting landslides and thus verify or adjust the machine learning based models and the factors used. On the basis of generated maps, Figure 10b,d,f,h, and Figure 12a are suggested to be used for landslide risk assessment because the susceptibility distributions are reliable as listed in Table 3, Table 4 and Table 6. Future studies can further include other information to achieve more accurate results of landslide risk assessments and management and to assist in land planning and policy making.

## 5. Conclusions

The development, which integrated the RF algorithm and cost-sensitive analysis with the GIS datasets and remotely sensed images, is presented to evaluate multi-temporal and event-based landslide susceptibilities for the Shimen reservoir watershed and is compared with three commonly used algorithms in the related field (i.e., DT, Bayes network, and logistical regression). The experimental results of space-robustness verification indicated that the RF algorithm outperformed the others, and all RF accuracies for the multi-temporal and event-based landslide susceptibility models were higher than 0.93 (Figure 3 and Figure 11). However, using such models to predict other events (time-robustness verification) may not produce plausible results because of the over-fitting issue. It is similar to the results of Rossi et al. [6], Tsai et al. [10] and Chang et al. [46].

To address the over-fitting issue and to improve the prediction capability, cost-sensitive analysis was conducted to adjust decision boundaries. The constructed multi-temporal and event-based models were evaluated and verified. The results in most cases indicated that the developed models produced better and more stable accuracies (Figure 6, Figure 7, Figure 8 and Figure 9 and Table 5). Regarding the unequal sample ratios in this study, reducing missing error was critical and the commission error was expected to decrease through further tasks in landslide risk assessment and management framework. The accuracies of the constructed models were greater than 0.75 in most cases for time-robustness verifications. To further examine the developed models, the landslide susceptibility distributions of true occurrence samples (Table 3, Table 4 and Table 6) and the generated landslide susceptibility maps (Figure 10 and Figure 12) were compared. They demonstrate that using cost-sensitive analysis can provide more reasonable results than the original algorithms. Furthermore, landslide susceptibility map generated through the results of the developed method revealed that the multi-temporal models were unaffected by the sample ratios, but use of extremely unbalanced sample ratios for event-based modeling was not suggested. 

The developed landslide susceptibility modeling framework also contributes to decision and policy making for better land planning and watershed management, and for better understanding of the eco-environmental impacts in the affected areas. In practice, collecting detailed field data is usually costly and time-consuming at a regional scale and may further reduce the effectiveness of knowledge-driven approaches. In such cases using geo-spatial technologies for data acquisition, processing and analysis may provide a more economic and efficient alternative for landslide assessments. On the other hand, in the knowledge-driven approaches, the stability can be determined accurately for specific slope sites, and these results are valuable for other cases. The outcomes of data-driven approaches are usually suitable for areas with long-term observation and data collection. Therefore, combining data- and knowledge-driven approaches for multi-scale landslide analysis and mapping can help support land planning and policy marking. Finally, the effect of the quality of landslide susceptibility models on the consequent tasks could be also examined in the future.

## Figures and Tables

**Figure 1 sensors-19-03717-f001:**
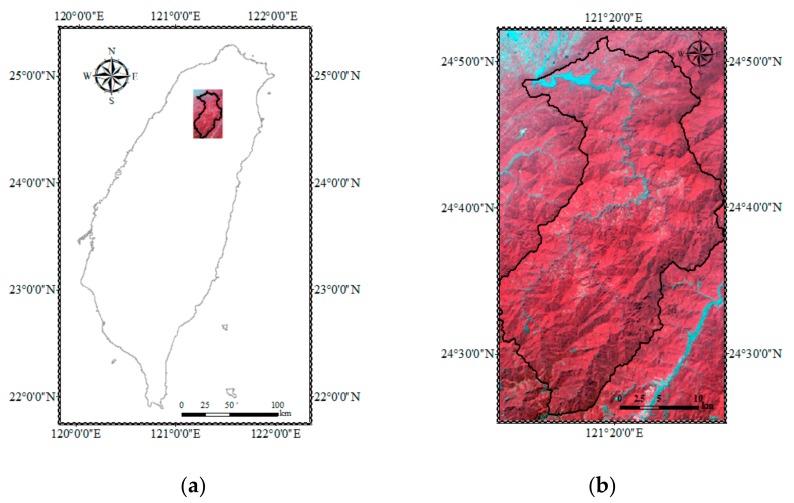
Study area and SPOT-5 satellite image. (**a**) Location with a false-color image, (**b**) zoomed-in image of the location.

**Figure 2 sensors-19-03717-f002:**
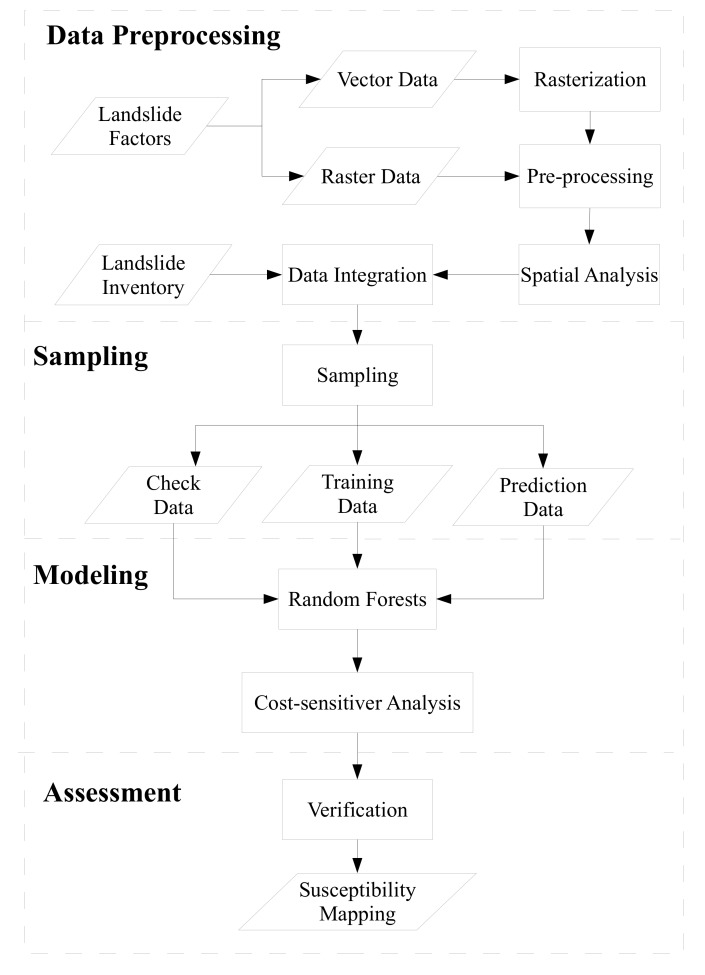
Procedure for the developed machine learning based landslide susceptibility assessments.

**Figure 3 sensors-19-03717-f003:**
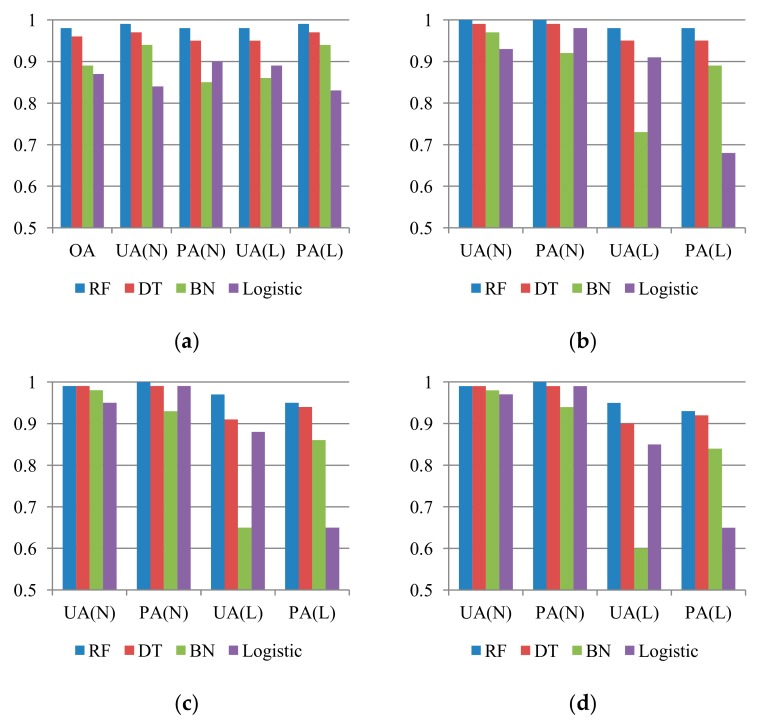
Space-robustness accuracies (*Y* axis) of multi-temporal landslide susceptibility assessments, considering different algorithms and proportions of landslide (L) and non-landslide (N) classes, where RF, DT, BN, Logistic, OA, UA, and PA represent random forests, decision tree, Bayes network algorithms, and logistic regression, overall accuracy, user’s accuracy, and producer’s accuracy, respectively. (**a**) L:N = 1:1; (**b**) L:N = 1:4; (**c**) L:N = 1:7; (**d**) L:N = 1:10.

**Figure 4 sensors-19-03717-f004:**
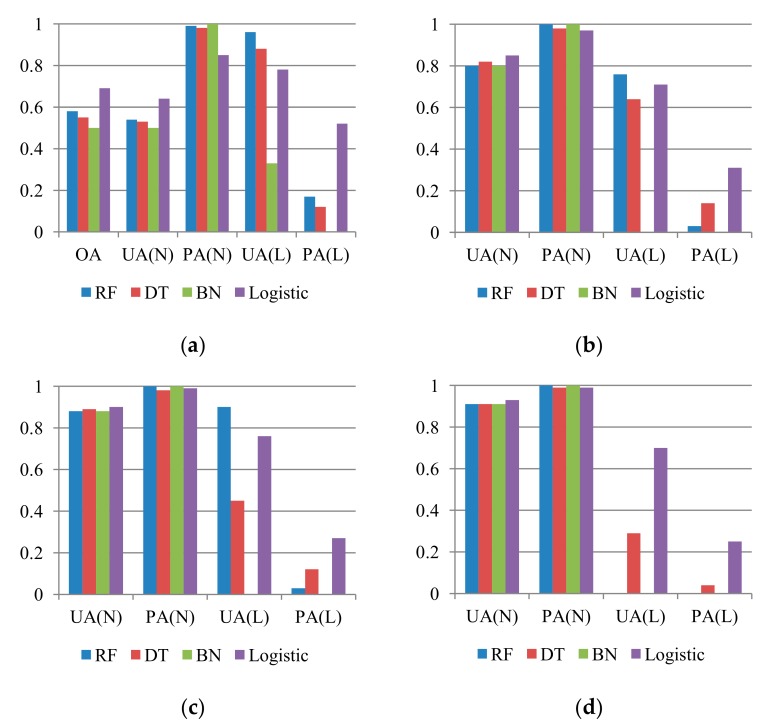
Time-robustness accuracies (*Y* axis) for predicting the 2008 dataset, considering different algorithms and proportions of landslide (L) and non-landslide (N) classes, where RF, DT, BN, logistic, OA, UA, and PA represent random forests, decision tree, Bayes network algorithms, logistic regression, overall accuracy, user’s accuracy, and producer’s accuracy, respectively. (**a**) L:N = 1:1; (**b**) L:N = 1:4; (**c**) L:N = 1:7; (**d**) L:N = 1:10.

**Figure 5 sensors-19-03717-f005:**
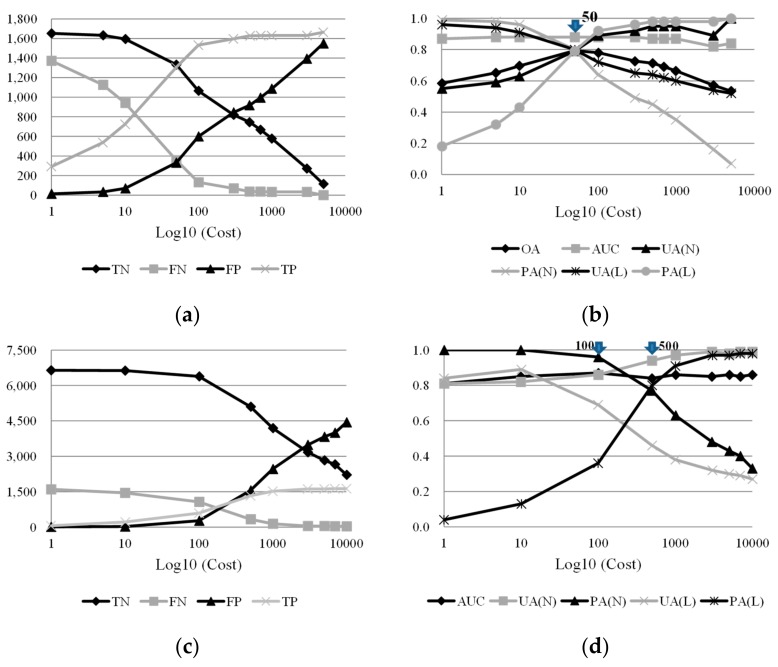
Counts in confusion matrices (left columns, *Y* axis: counts) and quantitative accuracies (right columns, *Y* axis: accuracies) of predicting the 2008 dataset using the RF algorithm with cost-sensitive analysis in different sample proportion cases, where TN, FN, FP, TP, AUC, L, and N indicate true negative, false negative, false positive, true positive, and area under ROC curve, landslide class, and non-landslide class, respectively. (**a**), (**b**) L:N = 1:1; (**c**), (**d**) L:N = 1:4; (**e**), (**f**) L:N = 1:7; (**g**), (**h**) L:N = 1:10.

**Figure 6 sensors-19-03717-f006:**
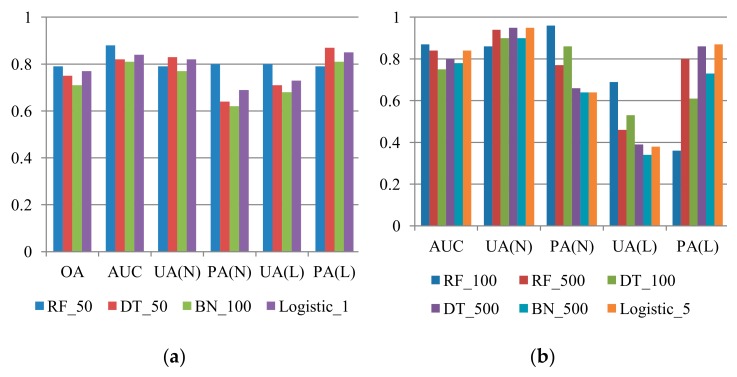
Comparison of the representative results (*Y* axis: accuracies) of 2008 dataset prediction obtained using RF, DT, BN, and logistic algorithm with cost-sensitive analysis, where RF, DT, BN,OA, UA, PA, AUC, L, and N indicate random forests, decision tree, and Bayes network algorithm, overall accuracy, user’s accuracy, producer’s accuracy, area under ROC curve, landslide class, and non-landslide class, respectively. (**a**) L:N = 1:1; (**b**) L:N = 1:4; (**c**) L:N = 1:7; (**d**) L:N = 1:10.

**Figure 7 sensors-19-03717-f007:**
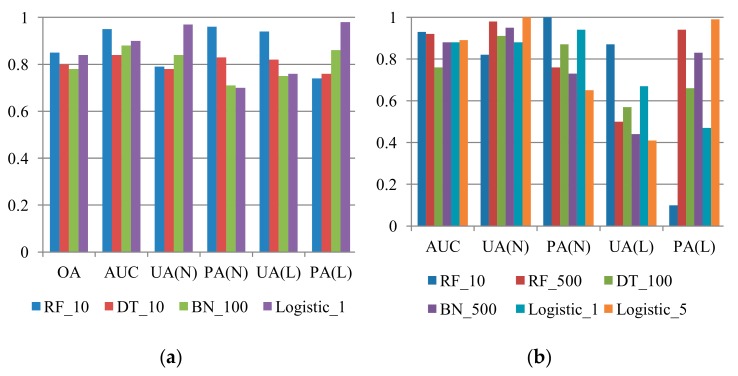
Comparison of the representative results (*Y* axis: accuracies) of Typhoon Fung-wong prediction using RF, DT, BN, and logistic algorithm with cost-sensitive analysis, where RF, DT, BN, OA, UA, PA, AUC, L, and N indicate random forests, decision tree, and Bayes network algorithm, overall accuracy, user’s accuracy, producer’s accuracy, area under ROC curve, landslide class, and non-landslide class, respectively. (**a**) L:N = 1:1; (**b**) L:N = 1:4; (**c**) L:N = 1:7; (**d**) L:N = 1:10.

**Figure 8 sensors-19-03717-f008:**
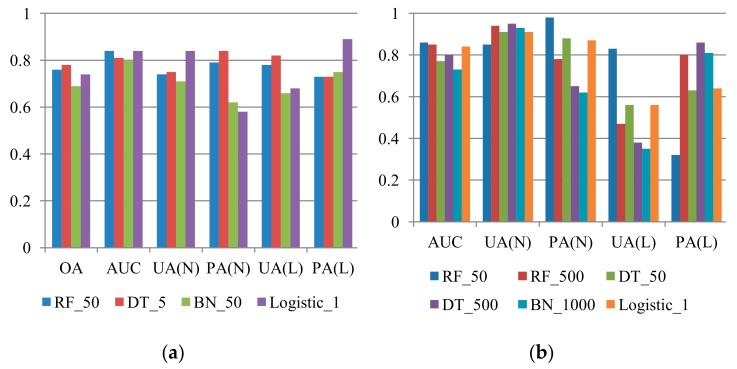
Comparison of the representative results (*Y* axis: accuracies) of Typhoon Sinlaku prediction using the RF, DT, BN, and logistic algorithm with cost-sensitive analysis, where RF, DT, BN, OA, UA, PA, AUC, L, and N indicate random forests, decision tree, and Bayes network algorithm, overall accuracy, user’s accuracy, producer’s accuracy, area under ROC curve, landslide class, and non-landslide class, respectively.(**a**) L:N = 1:1; (**b**) L:N = 1:4; (**c**) L:N = 1:7; (**d**) L:N = 1:10.

**Figure 9 sensors-19-03717-f009:**
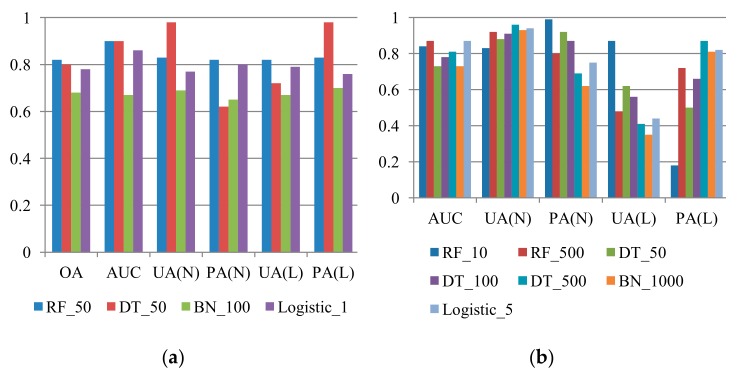
Comparison of the representative results (*Y* axis: accuracies) of Typhoon Jangmi prediction using the RF, DT, BN, and logistic algorithm with cost-sensitive analysis, where RF, DT, BN, OA, UA, PA, AUC, L, and N indicate random forests, decision tree, and Bayes network algorithm, overall accuracy, user’s accuracy, producer’s accuracy, area under ROC curve, landslide class, and non-landslide class, respectively.(**a**) L:N = 1:1; (**b**) L:N = 1:4; (**c**) L:N = 1:7; (**d**) L:N = 1:10.

**Figure 10 sensors-19-03717-f010:**
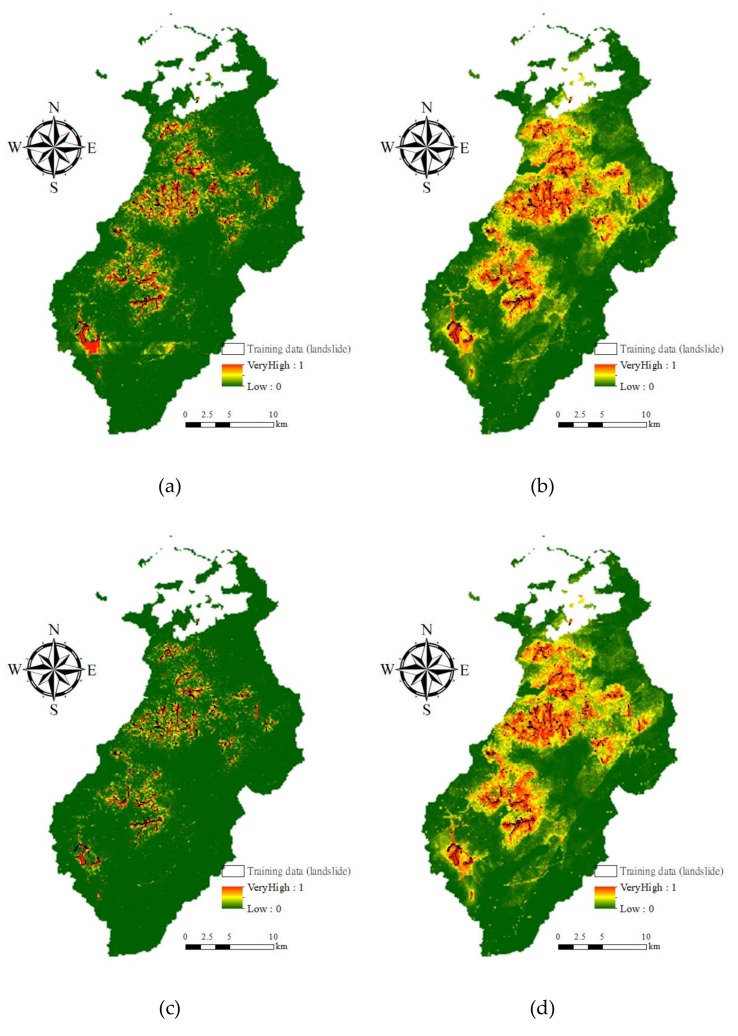
The multi-temporal landslide susceptibility maps generated (**a**) without cost analysis in the 1:1 case,(**b**) with cost = 50 in the 1:1 case, (**c**) without cost analysis in the 1:4 case, (**d**) with cost = 500 in the 1:4 case, (**e**) without cost analysis in the 1:7 case, (**f**) with cost = 1000 in the 1:7 case, (**g**) without cost analysis in the 1:10 case, and (**h**) with cost = 3000 in the 1:10 case.

**Figure 11 sensors-19-03717-f011:**
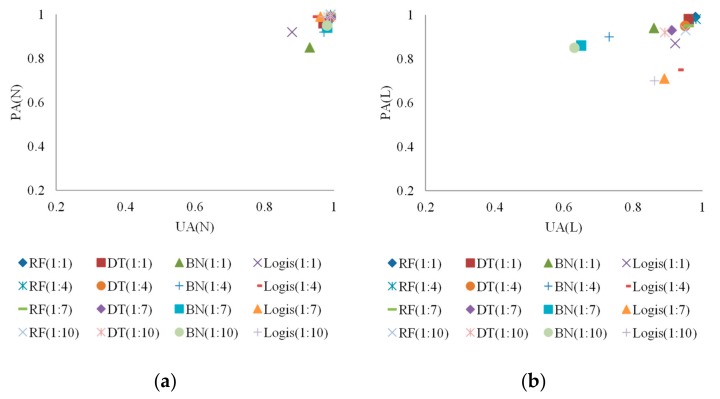
Space-robustness accuracies (*Y* axis) of event-based landslide susceptibility assessments, considering different algorithms and proportions of landslide and non-landslide classes, where RF, DT, BN, and logistic indicate random forests, decision tree, Bayes network algorithm, and logistic algorithm, respectively. (**a**) Non-landslide detections for Typhoon Aere; (**b**) Landslide detections for Typhoon Aere; (**c**) Non-landslide detections for Typhoon Matsa; (**d**) Landslide detections for Typhoon Matsa; (**e**) Non-landslide detections for Typhoon Sinlaku; (**f**) Landslide detections for Typhoon Sinlaku.

**Figure 12 sensors-19-03717-f012:**
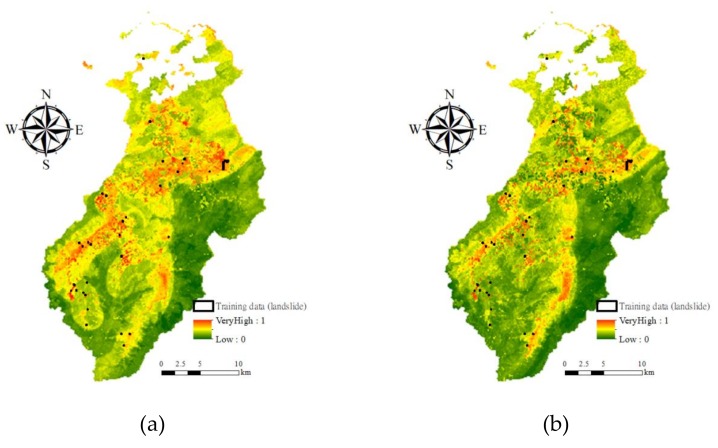
The landslide susceptibility maps generated for Typhoon Sinlaku (**a**) with cost = 50 in the 1:1 case, (**b**) with cost = 300 in the 1:4 case, (**c**) with cost = 500 in the 1:7 case, and (**d**) with cost = 700 in the 1:10 case.

**Table 1 sensors-19-03717-t001:** Typhoon events and the number of landslide samples used from the landslide inventory.

Typhoon	Date	Accumulated Precipitation (mm)	No. of Landslide Samples
Mindulle	2004/7/1–7/2	57–188	184
Aere	2004/8/25–8/26	85–1100	32,020
Nock-ten	2004/10/25	119–399	<50
Haitang	2005/7/18–7/20	278–799	100
Matsa	2005/8/4–8/5	188–636	2,112
Talim	2005/9/1	135–535	201
Bilis	2006/7/13–7/15	44–151	<50
Kaemi	2006/7/24	6–96	109
Bopha	2006/8/9	9–108	<50
Sepat	2007/8/16–8/19	38–340	331
Wipha	2007/9/18–9/19	104–360	309
Korsa	2007/10/6–10/7	265–512	430
Kalmaegi	2008/7/17–7/18	265–512	83
Fung-wong	2008/7/28–7/19	65–251	330
Sinlaku	2008/9/14–9/15	235–515	943
Jangmi	2008/9/28–9/29	258–473	308

**Table 2 sensors-19-03717-t002:** Landslide factors used in the study.

Original Data	Original Resolution/Scale	Used Factor (Raster Format)
DEM	20 m × 20 m	Aspect
		Curvature
		Elevation
		Slope
Geology map	1/50,000	Geology
Land-cover map	1/5000	Land-cover
Soil map	1/25,000	Soil
Fault map	1/50,000	Distance to fault
Rainfall-gage		Accumulative hourly rainfall maps (IDW)
		Accumulative hourly rainfall maps (Kriging)
		Maximum hourly rainfall maps (IDW)
River map	1/5000	Distance to river
Road map	1/5000	Distance to road
Satellite imagery	10 m × 10 m	NDVI

Abbreviations: IDW, inverse distance weighting; NDVI, normalized difference vegetation index.

**Table 3 sensors-19-03717-t003:** Landslide susceptibility distributions (Unit: %) of the 2008 occurrence samples based on the random forest algorithm with cost-sensitive analysis.

L:N	Cost	Low	Medium to Low	Medium to High	High	Very High
1:1	1	42.0	28.5	21.0	8.2	0.3
	50	2.9	6.7	26.9	**26.7**	**36.8**
1:4	1	71.4	15.6	11.6	1.4	0.0
	500	2.7	7.8	27.7	**26.5**	**35.3**
1:7	1	73.1	18.9	7.1	0.9	0
	1000	3.6	14.5	25.4	**26.3**	**30.2**
1:10	1	79.1	16.2	4.7	0.0	0.0
	3000	2	2.8	12.5	**32.6**	**50.1**

Abbreviations: L, landslide class; N, non-landslide class.

**Table 4 sensors-19-03717-t004:** Landslide susceptibility distributions (Unit: %) of the single-event occurrence samples based on the RF algorithm with cost-sensitive analysis.

Typhoon	L:N	Cost	Low	Medium to Low	Medium to High	High	Very High
Fung-wong	1:1	10	0	9.1	54.2	**29.1**	**7.6**
	1:4	500	0	3.6	14.2	**39.4**	**42.8**
	1:7	500	0	7.2	56.4	**15.8**	**20.6**
	1:10	1000	0	10.9	34.2	**33.9**	**21**
Sinlaku	1:1	50	5.2	8.8	33.1	**20**	**32.9**
	1:4	500	4.9	8.1	30.5	**22.4**	**34.1**
	1:7	3000	3.5	5.1	14.5	**34.4**	**42.5**
	1:10	3000	5.7	5.4	32.4	**22.8**	**33.7**
Jangmi	1:1	50	0	9.1	23.4	**27.6**	**39.9**
	1:4	500	0	13	24.7	**25**	**37.3**
	1:7	3000	0	0	18.5	**22.7**	**58.8**
	1:10	3000	0	8.7	28.9	**23.4**	**39**

Abbreviations: L, landslide class; N, non-landslide class.

**Table 5 sensors-19-03717-t005:** Best results of the event-based landslide susceptibility assessments.

P	L:N	T	Method.	Cost	OA (%)	AUC	UA (N)	PA (N)	UA (L)	PA (L)
Matsa	1:1	Sinlaku	RF	50	73.48	0.83	0.72	0.77	0.75	0.70
	1:4		BN	500		0.82	0.98	0.71	0.45	0.95
	1:7		DT	50		0.78	0.95	0.94	0.59	0.63
	1:10		DT	100		0.78	0.96	0.93	0.48	0.65
Sinlaku	1:1	Aere	Logistic	5	76.62	0.8	0.78	0.74	0.75	0.79
	1:4		RF	1000		0.86	0.90	0.86	0.53	0.63
	1:7		RF	3000		0.86	0.95	0.88	0.43	0.65
	1:10		RF	5000		0.84	0.96	0.89	0.36	0.60
Aere	1:1	Sinlaku	RF	50	83.43	0.9	0.8	0.89	0.88	0.78
	1:4		RF	300		0.89	0.92	0.95	0.78	0.68
	1:7		Logistic	70000		0.92	0.96	0.96	0.74	0.73
	1:10		RF	700		0.89	0.97	0.96	0.65	0.66

Abbreviations: AUC, area under ROC curve; BN, Bayes network algorithm, DT, decision tree; L, landslide class; N, non-landslide class; OA, overall accuracy; P, prediction events; PA, producer’s accuracy; RF, random forests; T, training events; UA, user’s accuracy.

**Table 6 sensors-19-03717-t006:** Landslide susceptibility distributions (Unit: %) of the event-based occurrence samples on the basis of RF algorithm with cost-sensitive analysis.

P	T	L:N	Cost	Low	Medium to Low	Medium to High	High	Very High
Matsa	Sinlaku	1:1	50	0	0.7	6.8	**83.1**	**9.4**
		1:4	700	0	1.7	34.6	**60.1**	**3.6**
		1:7	3000	0	0.2	28.1	**50**	**21.7**
		1:10	3000	0	8.2	64.5	**20.8**	**6.5**
Sinlaku	Aere	1:1	100	5.1	13.9	29	**27**	**25.1**
		1:4	1000	4.7	17.6	27.6	**25.1**	**25**
		1:7	3000	5.7	17.3	22.7	**25**	**29.3**
		1:10	5000	8.1	20.3	28.7	**19.3**	**23.6**
Aere	Sinlaku	1:1	50	1.8	11.4	23.8	**62.5**	**0.5**
		1:4	300	6	19.3	26.5	**48.2**	**0**
		1:7	500	10.2	14.4	17.9	**56.4**	**1.1**
		1:10	700	16.5	8.8	21.4	**52.7**	**0.6**

Abbreviations: L, landslide class; N, non-landslide class; P, prediction events; T, training events.

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
