# Peer review of "Improving GIS-Based Landslide Susceptibility Assessments with Multi-temporal Remote Sensing and Machine Learning"

_sensors, 2019, doi:10.3390/s19173717_

Round 1

Reviewer 1 Report

General comment:

This manuscript employs and compares random forest algorythm-based landslide susceptibility mapping with three other approaches and documents good performance of the suggested method. I’m impressed by the amount of work done and presented. Structure and the language are clear and understandable and I have no major objection against this study, just few moderate and minor suggestions.

As for the moderate suggestions, I recommend: (i) introducing a figure illustrating the whole procedure (since the methodology has many steps, that would surely be beneficial for readers); (ii) many landslide susceptibility maps are presented, using different combinations of parameters (Fig. 9 and 11); at the same time, the authors state that „maps can further be used for landslide risk assessment“ (L491-492) - which one of the produced maps would you recommend for practitioners and why?

Minor comments:

L38: what are these tasks and where is susceptibility mapping in this framework?

L54: what are high-dimensional data?

L58: please explain this proportion here, where it is mentioned for the first time

L64-67: please consider moving to methods section

L78-79: strategies or methods?

L82-83: studies or results are less reliable? Please check

L87-88: please provide examples / references of these studies

L99: according to Tab. 1 there are thousands of landslides; I suggest to replace „several“ by „numerous“ or similar

L102-104: does that mean that 31% has slope > 28°

L113: delete „are“

L136-138: not necessary if references above

L143: replace „observed“ by „developed“, „employed“ or similar

L146: time-robustness

L188-189: this is repeating at several places in the manuscript, please eliminate where not necessary

L218-221: better fitting in methods?

L267-271: these possible explanations might be better fitting in discussion

Fig. 5: greyscale (to be graphically unified with Figs. 2 and 3)? The same for Figs. 6-8

Tabs. 3-6: Higher share in high and very high class indicates better performance, correct? If so, please consider highlighting the best-performing combination

Figs. 9 and 11: what are the whitish parts in the north of the study area?

L441-443: this is repeating at several places in the manuscript, please eliminate where not necessary

Reviewer 2 Report

See attached file.

Reviewer 3 Report

Dear authors,

thank you for your study dealing a methodology which integrates the RF algorithm and cost-sensitive analysis with the GIS datasets and remote sensing to evaluate multi-temporal and event-based landslide susceptibility. However, there are some aspects related to the aim of the manuscript not discussed and/or presented properly. I would like to request you to consider the following comments in the revised version of the manuscript. 

In the 2.1 Study Site and Data Preprocessing section the description of the study area is quite short and synthetic. I would advise you to extend it by describing the main features of Shimen reservoir watershed in terms of morphology, geology and geomorphology to better contextualize the study area. Moreover, there are no references to the source of the DEM data from which the elevation was measured.

Figure 1. Reorganize it in a clearer way so that the readers very clearly see the location of the study area. Moreover, there are no references to the source of the SPOT-5 satellite images.

Line 167: Insert the reference to the equations (1), (2), (3), (4), (5).

Line 208: Insert the reference to the equations (11), (12), (13).

Figure 4: Re-organize it in a clearer way to depict it in a more complete and orderly manner. I would advise you to re-organize the layout in a single figure in order to place it on one page.

I suggest making the same re-organization also to Figure 5, 6, 7 and 8.

Figure 9: Re-organize it in a clearer way. According to this layout, it is not easy to locate the training data (landslide).
